# Incidental Detection of Classical Galactosemia through Newborn Screening for Phenylketonuria: A 10-Year Retrospective Audit to Determine the Efficacy of This Approach

**DOI:** 10.3390/ijns10010002

**Published:** 2023-12-22

**Authors:** Nathan W. P. Cantley, Robert Barski, Helena Kemp, Sarah L. Hogg, Hoi Yee Teresa Wu, Ann Bowron, Catherine Collingwood, Jennifer Cundick, Claire Hart, Lynette Shakespeare, Mary Anne Preece, Helen Aitkenhead, Sarah Smith, Rachel S. Carling, Stuart J. Moat

**Affiliations:** 1South West Newborn Screening and Metabolic Laboratory, Severn Pathology, Southmead Hospital, Bristol BS10 5NB, UK; 2Biochemical Genetics, Specialist Laboratory Medicine, Block 46, St James University Hospital, Leeds LS9 7TF, UK; robert.barski@nhs.net; 3Biochemical Genetics Unit, Cambridge University Hospitals, Cambridge CB2 0QQ, UK; 4Willink Biochemical Genetics Laboratory, Genomic Medicine, Manchester University NHS Foundation Trust, Manchester M13 9WL, UK; 5Metabolic and Newborn Screening Section, Department of Blood Sciences, Newcastle upon Tyne Hospitals NHS Foundation Trust, Newcastle-Upon-Tyne NE1 4LP, UK; 6Biochemistry Department, Alder Hey Children’s NHS Foundation Trust, Liverpool L12 2AP, UK; 7Regional Newborn Screening Laboratory, Royal Victoria Hospital, Belfast BT12 6BA, UK; 8Department of Clinical Chemistry and Newborn Screening, Sheffield Children’s Hospital, Sheffield S10 2TH, UK; 9Newborn Screening and Biochemical Genetics, Paediatric Laboratory Medicine, Birmingham Children’s Hospital, Steelhouse Lane, Birmingham B4 6NH, UK; 10Department of Chemical Pathology, Great Ormond Street Hospital for Children, London WC1N 3JH, UK; 11Scottish Newborn Screening Laboratory, Queen Elizabeth University Hospital, Glasgow G51 4TF, UK; 12Biochemical Sciences, Synnovis, Guys & St Thomas’ NHS Foundation Trust, London SE1 7EH, UK; 13GKT School Medical Education, Kings College London, London WC2R 2LS, UK; 14Department of Medical Biochemistry, Immunology & Toxicology, University Hospital Wales, Cardiff CF14 4XW, UK; stuart.moat@wales.nhs.uk; 15School of Medicine, Cardiff University, Cardiff CF14 4XW, UK

**Keywords:** classical galactosaemia, newborn screening, phenylketonuria, phenylalanine, tyrosine

## Abstract

In the UK, Classical Galactosaemia (CG) is identified incidentally from the Newborn Screening (NBS) for phenylketonuria (PKU) using an “Other disorder suspected” (ODS) pathway when phenylalanine (Phe) and tyrosine (Tyr) concentrations are increased. We aimed to determine the efficacy of CG detection via NBS and estimate the incidence of CG in live births in the UK. A survey was sent to all UK NBS laboratories to collate CG cases diagnosed in the UK from 2010 to 2020. Cases of CG diagnosed were determined if detected clinically, NBS, or by family screening, as well as age at diagnosis. Cases referred via the ODS pathway were also collated, including the final diagnosis made. Responses were obtained from 13/16 laboratories. Between 2010 and 2020, a total of 6,642,787 babies were screened, and 172 cases of CG were identified. It should be noted that 85/172 presented clinically, 52/172 were identified by NBS, and 17/172 came from family screening. A total of 117 referrals were made via the ODS pathway, and 45/117 were subsequently diagnosed with CG. Median (interquartile range) age at diagnosis by NBS and clinically was 8 days (7–11) and 10 days (7–16), respectively (Mann–Whitney U test, U = 836.5, *p*-value = 0.082). The incidence of CG is 1:38,621 live births. The incidence of CG in the UK is comparable with that of other European/western countries. No statistical difference was seen in the timing of diagnosis between NBS and clinical presentation based on the current practice of sampling on day 5. Bringing forward the day of NBS sampling to day 3 would increase the proportion diagnosed with CG by NBS from 52/172 (30.2%) to 66/172 (38.4%).

## 1. Introduction

Newborn screening (NBS) for phenylketonuria (PKU) in the United Kingdom (UK) began in the late 1950s and since 1969 has relied upon the measurement of phenylalanine (Phe) in whole blood applied to filter paper collected on days 5 to 8 of life [1]. The addition of tyrosine (Tyr) as a second-line test as part of the PKU screening protocol from 2005 has improved the specificity in the detection of cases of PKU, as well as delineating other possible causes [2]. An elevated Phe concentration can be found secondary to other medical conditions, such as liver dysfunction and inherited metabolic disorders (IMD), including Classical Galactosaemia (CG) [3].

CG has been screened for as part of newborn screening programmes internationally since 1964. Direct screening tests for CG are used, including an enzyme activity test of Galactose-1-Phosphate-Uridyltransferase (GALT) and/or the measurement of total blood galactose concentration [4]. The United States of America (USA), with a CG incidence of 1:50,000 live births, has been screening for CG since the early 1960s and uses the GALT enzyme method [5]. In the Republic of Ireland (RoI), the CG incidence is quoted as 1:19,000 infants born, though this is heavily skewed by the 1:450 births found within the Traveller community (the non-traveller Irish community incidence is 1:36,000 births) [6]. CG screening is offered to all newborns from the traveller community in RoI. Importantly, observational data published by the Galactosaemia Network (GalNet) in 2019, which included patient cases from 15 countries (including the UK), found that those individuals identified by NBS had a lower odds ratio for neonatal complications due to CG compared to those identified by other means (OR 0.30, 95% confidence intervals 0.20–0.47, *p* < 0.001) [7]. However, there remains significant controversy and divergence in opinion on the benefits of NBS for CG.

The UK National Screening Committee (NSC) does not recommend screening for CG due to a lack of high-quality published evidence to support its inclusion, citing the following reasons [8]:Affected individuals show symptoms of galactosaemia at an average of 7 days of age—this is equal to or quicker than the point at which NBS results are available;An accurate screening test is not available and carries an unacceptable risk of misdiagnosis (false positive rate)—including identifying cases of uncertain clinical significance;It is unclear if early treatment in diagnosed individuals changes the long-term outcome (despite the observational evidence apparent from the GalNet registry).

Thus, within the UK, cases of CG by NBS are detected incidentally based on high Phe and Tyr identified from screening for PKU. Referrals to paediatric IMD services are advised using an “other disorder suspected” (ODS) clinical referral pathway (Figure 1) [9]. An urgent galactosaemia test in the form of a galactosaemia screen (GALT enzyme activity), thin-layer chromatography of sugars, or measurement of Galactose-1-Phosphate is recommended. However, laboratory practice is not harmonised across the UK in implementing this. The number of incidental cases of CG detected through NBS in the UK has not previously been evaluated. Furthermore, the most recently published data on the incidence of CG in UK newborns dates to a pilot study in 1990, where the incidence was estimated at 1:44,000.

This study sought to:assess the efficacy of identifying incidental cases of CG within UK NBS laboratories using the current ODS pathway;determine the proportion of individuals with CG diagnosed in the UK through the NBS pathway as compared to clinical presentation;provide an updated estimate of CG incidence in UK live births.

## 2. Materials and Methods

NBS laboratories were asked to collate data from 1 June 2010 to 31 May 2020 from two perspectives:number of NBS referrals made using the ODS pathway from PKU screening and the final diagnosis from that referral, where known;number of cases of CG diagnosed by the laboratory and the mechanism by which they were diagnosed (e.g., clinical presentation vs. NBS).

The cases collated were of Classical Galactosaemia and not of Duarte or rare variant sub-types of Galactosaemia. No patient-sensitive information was collected as part of the survey. Duplicate cases of CG submitted by different centres were excluded from the final data analysis by the provision of the last four digits of the NHS number. For assessing the outcomes of the ODS pathway, cases were excluded from the data analysis if laboratories were not using the recommended referral pathway. Final diagnoses from cases identified through the ODS pathway were stratified into the following groups:CG;other IMD;liver disease/liver immaturity;not known/not stated;

For assessing how CG was identified, cases were stratified into the following groups:clinically presenting;NBS;pre-symptomatic testing due to a family history of CG;not known.

Where available, data on the age of diagnosis and the Phe and Tyr concentration on NBS were collected for cases of CG, irrespective of how cases were identified, and descriptive statistics (median and interquartile range) were evaluated for both parameters. Statistical analysis of the difference between groups in the median age of diagnosis was assessed using Mann–Whitney U at a significance level of 0.05, given the non-parametric nature of the data collected. Linear regression was used to identify a correlation between biochemical data and age at the time of diagnosis. The total number of babies screened through each NBS laboratory over the data collection period was collected and used to calculate the UK CG incidence rate using the total number of CG cases diagnosed in the UK over the study period. The total number of babies born in the study period was collated from published data by the Office for National Statistics (ONS) [10]. Ethical approval for the study was not sought as this information was collected within the context of service improvement. Data were stored in accordance with data protection guidelines. This observational study is reported in line with STROBE guidelines [11].

## 3. Results

Responses were received from 13/16 (81.25%) of UK NBS laboratories. 10/13 NBS laboratories in England responded, in addition to each of the devolved nation NBS laboratories in Scotland, Wales, and Northern Ireland (all UK nations follow a common NBS protocol).

### 3.1. Incidental Diagnosis of CG from ODS PKU Pathway

A total of 117 referrals were made through the ODS section of the PKU pathway in the study period across all responding labs. Of these, 45/117 (38.5%) were subsequently diagnosed with CG, 22/117 (18.8%) were due to underlying liver disease/dysfunction, 8/117 (6.8%) were subsequently diagnosed with an IMD other than CG (this included 4/8 diagnosed with Citrin deficiency and 1/8 diagnosed with B12 non-responsive Methylmalonic Aciduria), and in 42/117 (35.9%) data on the final diagnosis were not available.

### 3.2. Pathway for Diagnosing Cases of CG Using PKU Screening Pathway and Incidence Calculation

A total of 172 cases of CG were diagnosed over the data collection period from UK laboratories. 85/172 (49.4%) presented clinically, 52/172 (30.2%) were detected prior to clinical presentation via the “other disorders suspected” section of the PKU NBS pathway, and 17/172 (9.9%) were diagnosed due to pre-symptomatic investigation due to family history. The details of how the cases were diagnosed were unknown in 18/172 (10.5%). Data on age at the time of diagnosis and a summary of Phe and Tyr concentrations at the time of diagnosis, where known, are shown in Table 1. No correlation by linear regression was seen between age at the time of diagnosis and Phe (R^2^ = 0.01) or Tyr (R^2^ = 0.06) concentrations, respectively. The median (interquartile range) day of diagnosis between those presenting clinically and those detected incidentally via NBS was not statistically different when tested by Mann–Whitney U (Clinical: 10 (7–16) days, NBS: 8 (7–11) days, U = 836.5, *p* = 0.082). A total of 6,642,787 babies were screened across all labs that responded out of 7,667,187 babies born in the UK across the 10-year data collection period (86.6%). Using a total case number of 172 and the total number of babies screened, the incidence rate for the UK was estimated to be 1:38,621.

## 4. Discussion

A central element presented against the inclusion of CG in NBS is that screening results for CG do not yield an earlier diagnosis, as cases are diagnosed clinically by day 7 of life. In our evaluation, 39% of CG cases in our review period were detected prior to clinical presentation, either incidentally via the NBS pathway for PKU or due to early testing because of a family history of the disorder. The median day of diagnosis between those presenting clinically and those detected incidentally via NBS was not statistically different when tested by Mann–Whitney U (median 10 days and 8 days, respectively, U = 836.5, *p* = 0.082). Our data, therefore, do not refute the conclusion of the UK NSC. However, a significant confounding factor in this conclusion is the timing of sampling for the NBS programme. This is later after birth compared to other national programmes such as Italy (days 2–3), RoI (days 3–5), or the USA (days 1–2) [12]. If the UK NBS programme was aligned with other Western/European countries in recommending the collection of NBS samples on an earlier day of life, this could shift diagnosing more CG cases prior to clinical presentation.

On reviewing the day of presentation in the clinically presenting cases, this was known in 80/85 (94.1%) of cases. If the day of NBS sampling was brought forward to day 3 of life, allowing a median duration of 3 days for sample transport and analysis, an additional 14/80 (17.5%) of CG cases would have been identified by NBS by day 6 of life. This would increase the proportion diagnosed with CG by NBS by bringing the day of sampling forward to day 3, from 52/172 (30.2%) to 66/172 (38.4%). Unfortunately, the incomplete biochemical data in 8/14 of these additional cases means the above estimation would only hold true if all cases met the biochemical criteria.

In total, 53 IMD cases were identified over the 10-year period using the ODS pathway, with the vast majority (45/53, 84.9%) being CG. Citrin deficiency was the only other disorder identified more than once (4/53, 7.5%). This yields a positive predictive value (PPV) for detecting an IMD using the ODS pathway at 45% and for detecting CG at 38%. However, the incomplete data on the final diagnosis for cases referred to the ODS pathway (35.9% of cases referred) means evaluations on clinical utility are incomplete. Reviewing data on the clinical utility of CG screening from international NBS programmes, large-scale retrospective evaluations from the Netherlands and Italy yielded PPV’s of 0.14% and 6.9%, respectively [13]. While this seems like poor evidence, the Dutch programme using a combination of total galactose concentrations and GALT enzyme activity yielded 100% sensitivity and a high specificity of >95%.

Significantly, this study provides an updated estimate of the incidence of CG in UK newborns of 1:39,000 live births, which is higher than the 1990 estimate of 1:44,000. It aligns the UK incidence with the Republic of Ireland’s 1:36,000 (non-travelling community) and suggests a higher incidence compared to other Western countries such as the USA (1:50,000).

### Limitations of This Study

The fact that not all NBS laboratories in the UK responded means this is an incomplete dataset and may account for discrepancies between the CG cases identified and the final diagnoses from the ODS pathway. Movers in and out of the various regions/the UK may also account for discrepancies. The non-response from some laboratories may also mean the estimate for the incidence of CG per live birth brought together from this survey may not be truly reflective. A lack of full biochemical data and age at diagnosis on all CG cases diagnosed does not allow for a true evaluation of the sensitivity/specificity of the PKU screening protocol and ODS pathway. The fact that a final diagnosis was not known in 42/117 referrals to the ODS pathway is a limitation of assessing the diagnostic efficacy of the ODS pathway. Data are not known on what proportion of these cases represent possible prematurity, neonatal death, or non-IMD, non-liver dysfunction causes of elevated bloodspot amino acid concentrations, such as total parenteral nutrition (TPN). This could overestimate the diagnostic efficacy of the ODS pathway in identifying CG cases. The fact that this study evaluates the ODS pathway using Phe and Tyr is also a limitation, as this does not parallel other screening protocols that use more specific screening methods.

## 5. Conclusions

The incidence of CG in live births in the UK, as of 2020, is estimated to be 1:39,000. Cases of CG are incidentally detected through the NBS PKU pathway in the UK, indicating a useful secondary outcome. However, in part due to the day of NBS specimen collection, many cases of CG diagnosed clinically in the UK are still prior to NBS results being available. However, given what is known from international data on clinical outcomes for individuals with CG identified from NBS, a significant proportion of individuals newly diagnosed with CG benefit from incidental case findings from NBS.

## Figures and Tables

**Figure 1 IJNS-10-00002-f001:**
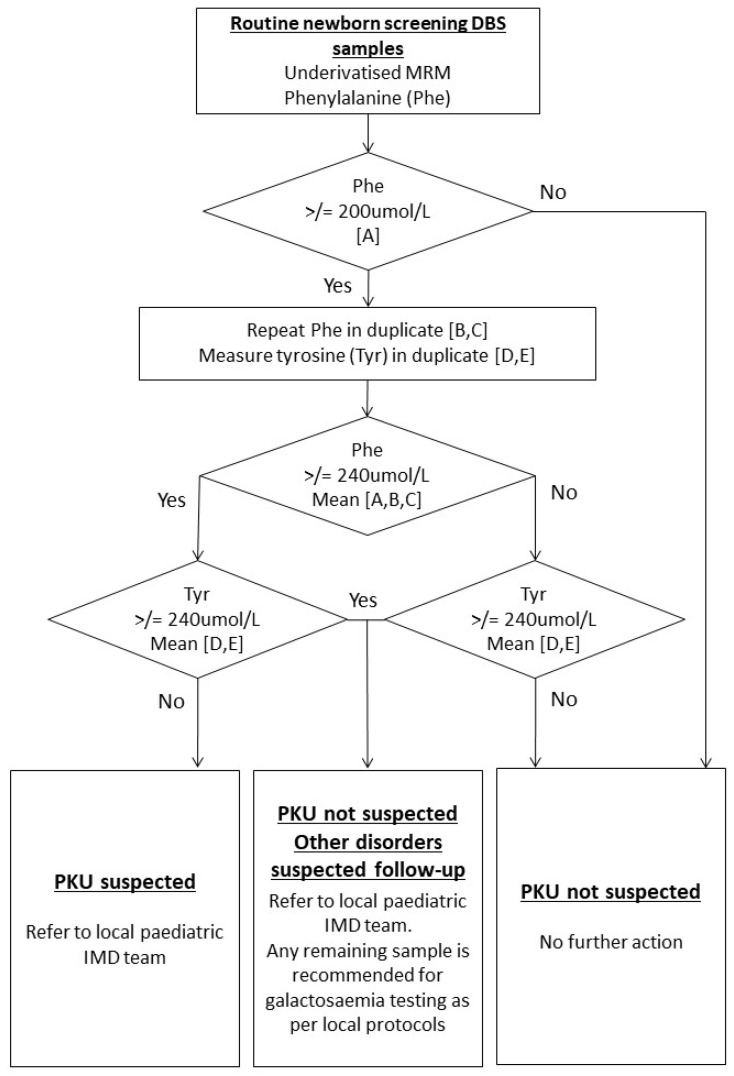
Current UK NBS protocol for the detection of suspected cases of Phenylketonuria (PKU) [9]. Figure contains public sector information licensed under the Open Government Licence v2.0.

**Table 1 IJNS-10-00002-t001:** Data showing the day of diagnosis and analyte concentrations at the time of diagnosis by method of case detected. Data are shown as the median (interquartile range) unless otherwise stated.

	Clinically Presenting	Newborn Screening (NBS)	Family
Number of cases (%) with data for the day of diagnosis	80/85 (94.1%)	27/52 (51.9%)	8/17 (47.1%)
Age of diagnosis—days	10 (7–16)	8 (7–11)	2 (1–4)
Number of cases with data for Phe and Tyr concentrations	45/85 (52.9%)	43/52 (82.7%)	10/17 (58.8%)
Phenylalanine concentration—μmol/L	150 (75–213)	305 (239–405)	73 (67–85)
Tyrosine concentration—μmol/L	371 (128–808)	878 (405–1072)	133 (103–177)

## Data Availability

The data presented in this study are available on request from the corresponding author.

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
