# Peer review of "Incidental Detection of Classical Galactosemia through Newborn Screening for Phenylketonuria: A 10-Year Retrospective Audit to Determine the Efficacy of This Approach"

_2409-515X, 2023, doi:10.3390/ijns10010002_

Round 1

Reviewer 1 Report

Comments and Suggestions for Authors

The date of sampling is out of date with current recommendations and the results are inevitably affected by this.

When referring to the period 2010-2020, the interest is less because it does not present the data of many of the GC cases.

The other types of Galactosemia and the Duarte forms are lost in this way and their interest is limited.

Despite this, it is true that about 30% of GC cases can be detected and in some countries they can be diagnosed early.

Author Response

Dear Reviewer,

Re: Manuscript ID: IJNS-2705887, Incidental detection of Classical Galactosemia through newborn screening for Phenylketonuria: A 10-year retrospective audit to determine the efficacy of this approach.

Thank you for your letter regarding the above entitled manuscript and for allowing me the opportunity to respond to your comments.

I will address the points in the order raised.

Reviewer 1

The date of sampling is out of date with current recommendations and the results are inevitably affected by this.

Response: The period of data collection (2010-2020) is in line with the current UK newborn screening programme for PKU with no changes to the protocol during or since the data collection period. The delay between data collection and presentation of this manuscript is multi-factorial, including delays due to COVID.

When referring to the period 2010-2020, the interest is less because it does not present the data of many of the GC cases.

Response: Data were provided from responding labs and our primary focus was to highlight the mechanism of diagnosis rather than draw correlations between cut-offs and diagnosis. The remaining case data are not able to be provided by respondents.  

The other types of Galactosemia and the Duarte forms are lost in this way and their interest is limited.

Response: Only cases of classical galactosaemia were reported and collated as part this study.  We have provided a statement in the methods section to clarify this fact (page 4, line 116).

Despite this, it is true that about 30% of GC cases can be detected and in some countries they can be diagnosed early.

Reviewer 2 Report

Comments and Suggestions for Authors

Comments and Suggestions for Authors

This manuscript presents the diagnosis pathways of classical galactosemia in the UK and compares the efficacy to other international approaches. Although there are important observations and comments in the Discussion, the averall scientific contribution of the study is moderate.

Questions

1.    Is there any explanation why the Phe and Tyr values were unavailable in some of the ODS cases?

2.    The 43 cases with Phe/Tyr available includes all 27 cases with age of diagnosis data? It would be nice to comment on how Phe/Tyr values change with age in newborns.

Comments

1.    Results: section 3.2. The number of newborns screened in the participating UK NBS laboratories is given. It would be informative to know the percentage/number of all babies born in the UK in this time period.

2.    The calculation of incidence in UK is slightly incorrect since not all newborns in UK are included.

Suggestions

It would give a good support for the scientific contribution of the study if the two groups would be compared based on clinical data of disease status after 1, 5 10 years.

Author Response

Dear Reviewer,

Re: Manuscript ID: IJNS-2705887, Incidental detection of Classical Galactosemia through newborn screening for Phenylketonuria: A 10-year retrospective audit to determine the efficacy of this approach.

Thank you for your letter regarding the above entitled manuscript and for allowing me the opportunity to respond to your comments.

I will address the points in the order raised.

Reviewer 2

Comments and Suggestions for Authors

This manuscript presents the diagnosis pathways of classical galactosemia in the UK and compares the efficacy to other international approaches. Although there are important observations and comments in the Discussion, the overall scientific contribution of the study is moderate.

Questions

  1. Is there any explanation why the Phe and Tyr values were unavailable in some of the ODS cases?

Response: Biochemical data were provided as a secondary outcome for cases collated by newborn screening laboratories and where available. We accept the absent biochemical data detracts from providing systematic conclusions on future practice should the day of sampling be brought forward. These missing data unfortunately cannot be provided.

  1. The 43 cases with Phe/Tyr available includes all 27 cases with age of diagnosis data? It would be nice to comment on how Phe/Tyr values change with age in newborns.

Response: Looking at linear regression plots of available data revealed no correlation between age at time of diagnosis and Phe/Tyr concentration. A statement clarifying this has been included in the results section of the manuscript (page 4, line 137 & page 5, line 165-166).

Comments

  1. Results: section 3.2. The number of newborns screened in the participating UK NBS laboratories is given. It would be informative to know the percentage/number of all babies born in the UK in this time period.

Response: Collating the number of live births recorded by the Office of National Statistics in the UK in the same time period – 6,399,704/7,667,187 (83.5%) of babies born in the UK are captured in the incidence calculation. We have added a statement in the results section highlighting this (Page 4, line 141).

  1. The calculation of incidence in UK is slightly incorrect since not all newborns in UK are included.

Response: Our study incorporated data from 13/16 newborn screening laboratories on 6.6M screened babies and highlights that our incidence calculation is an estimation, which we qualify in our manuscript (page 5, line 169-172). The included data is of sufficient proportion of the total babies screened to be able to derive externally valid estimations of CG incidence. 

Suggestions

It would give a good support for the scientific contribution of the study if the two groups would be compared based on clinical data of disease status after 1, 5 10 years.

Response: Data were collated at a single time point (time of diagnosis) without longitudinal clinical outcome data on disease status. We agree this would provide additional benefit. However, this is outside the scope of this cross-sectional survey.

Round 2

Reviewer 2 Report

Comments and Suggestions for Authors

Answers accepted

Author Response

Dear Reviewer,

Many thanks for your time in re-reviewing our manuscript and I am grateful for your time in confirming the answers to your questions were acceptable. 

Yours Sincerely, 

Dr Nathan Cantley

on behalf of all authors